# Quantifying the Structure and Properties of Nanomagnetic Iron Oxide Particles for Enhanced Functionality through Chemical Synthesis

**DOI:** 10.3390/nano13152242

**Published:** 2023-08-03

**Authors:** Johar Amin Ahmed Abdullah, Álvaro Díaz-García, Jia Yan Law, Alberto Romero, Victorino Franco, Antonio Guerrero

**Affiliations:** 1Departamento de Ingeniería Química, Escuela Politécnica Superior, Universidad de Sevilla, 41011 Sevilla, Spain; 2Departamento de Física de Materia Condensada, ICMS-CSIC, Universidad de Sevilla, 41012 Sevilla, Spain; adiaz18@us.es (Á.D.-G.); jylaw@us.es (J.Y.L.); vfranco@us.es (V.F.); 3Departamento de Ingeniería Química, Facultad de Química, Universidad de Sevilla, 41012 Sevilla, Spain; alromero@us.es

**Keywords:** nanoparticles, nanocomposite, iron oxide, chemical synthesis, stability, magnetic properties

## Abstract

This comprehensive study investigates the properties of chemical nanomagnetic iron oxide particles (CNMIOPs) synthesized through a chemical method. The primary objective is to examine how pH levels and washing solvents affect the magnetism properties of these nanoparticles. Three different pH levels (1.2, 7.5, and 12.5) using NaOH and two washing solvents (ethanol and water) are employed. The characterization techniques include FTIR, SEM, TEM, XRD, ZSP, and VSM. Furthermore, the study incorporates two specific pH- and solvent-dependent CNMIOPs into PCL electrospun materials to analyze their performance in a targeted application. The results show that pH and the washing process significantly affect the CNMIOPs’ properties. Higher pH levels result in smaller particles with higher crystallinity and reduce crystalline anisotropy. SEM and TEM analysis confirm different morphologies, including cubic, spherical, and elongated shapes. Ethanol-washed CNMIOPs exhibit superior magnetic behavior, with the highest magnetization saturation at pH 12.5 (Ms = 58.3 emu/g). The stability of the CNMIOPs ranges from −14.7 to −23.8 mV, and higher pH levels exhibit promising antioxidant activity. Furthermore, the study explores the effects of pH and washing solvents on CNMIOP-infused nanofiber membranes, with better dispersion observed with ethanol washing. Overall, this research provides valuable insights into the properties and behavior of CNMIOPs under varying pH and washing conditions.

## 1. Introduction

Nanomaterials have revolutionized various scientific fields because of their unique properties and potential applications [1]. Among them, nanomagnetic iron oxide particles (NMIOPs) have garnered significant attention for their exceptional magnetic properties and versatility in diverse technological areas [2]. The structural characteristics and properties of NMIOPs play a crucial role in determining their functionalities and applications. One pivotal area of interest lies in their magnetic properties, specifically focusing on saturation magnetization (Ms), coercivity (Hc), and remanence (Mr) [3]. Magnetite (Fe_3_O_4_), hematite (Fe_2_O_3_), and maghemite (γ–Fe_2_O_3_) are common examples of iron-derived oxides that exhibit superparamagnetic behavior when their size is below 20 nm. These nanoparticles demonstrate a lack of magnetization in the absence of an external magnetic field, rendering them extremely valuable for applications that demand the precise manipulation of their magnetic properties [4]. In addition to their magnetic properties, the high surface-to-volume ratio and nanoscale dimensions of NMIOPs provide them with enhanced binding capacity and stability in solution. They are biocompatible, non-toxic, and possess unique characteristics that have led to their widespread use in various fields. Some notable applications of NMIOPs include biomedicine, environmental remediation, catalysis, aerospace and defense, electronics, healthcare, construction, textiles, food industry, and agriculture [5].

However, several factors influence the magnetic properties, including the synthesis approach, coating techniques, and sample preparation. The atomic or molecular structure, electron spin, magnetic domains, temperature, applied magnetic field, composition, impurities, and crystal structure also play significant roles in determining the magnetic behavior of these nanoparticles. Efforts have been focused on developing simple and cost-effective synthesis approaches. Chemical methods are considered the simplest for controlling the size, shape, and surface properties of nanoparticles, which affect their physical and chemical characteristics [6]. However, some drawbacks related to purity, dispersity, crystalline structure, and tunable size can be controlled through parameters like composition and the processing solution [7]. Preventing agglomeration and ensuring compatibility with living systems require controlling particle size through comprehensive characterization and performance [8,9]. pH, the washing process, and the choice of washing solvent significantly influence the final properties of nanoparticles. However, there is still a lack of comprehensive investigation into the relationship between pH, washing solvent, and the resulting structure and properties of nanoparticles [10]. This study aims to fill this knowledge gap by quantifying the structure and properties of nanoparticles and exploring how these characteristics can be enhanced through chemical synthesis techniques. The influence of pH and washing solvent on the final properties will also be examined. A thorough understanding of these relationships facilitates the optimization of the synthesis process and the customization of nanoparticles for specific applications.

The primary objective of this study is to investigate how pH levels and washing solvents influence the magnetic properties of chemical nanomagnetic iron particles (CNMIOPs). To achieve this, three different pH levels (1.2, 7.5, and 12.5) using NaOH and two washing solvents (ethanol and water) were utilized. The study employed a range of characterization techniques, including FTIR, SEM, TEM, XRD, ZSP, and VSM, to provide comprehensive insights and accurate quantification during the preparation of nanoparticles. In addition, two specific pH- and solvent-dependent CNMIOPs were incorporated into electrospun PCL materials to assess their performance in a targeted application. Moreover, the study explores the effects of pH and washing solvents on CNMIOP-infused nanofiber membranes, including dispersion, aggregation, and functional properties. Overall, this research offers valuable insights into the properties and behavior of CNMIOPs under different pH and washing conditions, providing detailed procedures for enhancing magnetic nanoparticle properties and exploring their potential applications.

## 2. Materials and Methods

### 2.1. Materials

We used FeCl_3_·6H_2_O (iron (III) chloride hexahydrate) with a purity of 98%, CH_3_OH (methanol); ethanol with a purity of 99.9%, NaOH; C_7_H_6_O_5_ (gallic acid); DPPH (2,2-diphenyl-1-picrylhydrazyl); and C_2_H_6_SO (anhydrous dimethyl sulfoxide) with a purity of 99.9%. Additionally, chloroform (Friendemann Schmidt, Parkwood, Australia), DMF ((CH_3_)2NC(O)H) (N,N-Dimethylformamide) (Merck, Darmstadt, Germany), and PCL (Polycaprolactone) (C_6_H_10_O_2_)_n_ with an Mn of 80,000 (Sigma Aldrich, Saint Louis, MI, USA) were employed. All other chemicals and reagents used were of analytical grade.

### 2.2. Nanoparticle Synthesis

CNMIOPs were synthesized based on previous methods with modifications [11]. In summary, the synthesis involved gradually adding 20 mL of a reducing agent (1 M NaOH) to a solution of 20 mL of 1 M iron chloride (FeCl_3_·6H_2_O) while adjusting the pH reaction to 1.2, 7.5, or 12.5 using either FeCl_3_·6H_2_O or 5 M NaOH solution. The resulting solutions were then transferred to beakers and heated on a hot plate at 50 °C while stirred for 2 h. Subsequently, the mixture was filtered using Whatman nº1 filter papers and washed with ethanol at least three times. Furthermore, CNMIOPs synthesized at pH = 7.5 underwent a separate washing step with distilled water to evaluate the effect of the washing solvent. This additional washing process ensured the removal of impurities and foreign particles suspended in the mixture. To further prepare the samples, they were pretreated by drying them in an oven at 100 °C for 8 h. This drying process facilitated the removal of excess moisture and ensured proper sample preparation. Subsequently, a final heat treatment was carried out at 500 °C for 5 h. This high-temperature treatment was employed to eliminate any remaining excess matter and impurities, resulting in the isolation of the nanoparticles for subsequent characterization. It is worth noting that this two-step nanoparticle treatment involved intense heat transmission through a regulated drying oven and a muffle, ensuring efficient drying, sterilization, and removal of organic substances and impurities.

### 2.3. Nanoparticle Characterization

To investigate the properties of the synthesized CNMIOPs in this study, various experimental techniques were employed [11,12]. Fourier transform infrared spectroscopy (FTIR) was utilized to analyze structural information, particularly the identification of Fe-O bonds in the fingerprint region (800–400 cm^−1^). Scanning electron microscopy (SEM) with a Zeiss EVO scanning electron microscope was employed to examine the nanoparticle morphology and size, and image analysis was performed using the ImageJ software(v1.53q, NIH, Bethesda, MD, USA). Transmission electron microscopy (TEM) using a Talos S200 microscope allowed for further characterization of nanoparticle morphology and size. X-ray diffraction (XRD) analysis with a Bruker D8 Advance A25 diffractometer confirmed the crystalline phase, crystal systems, size, and degree of crystallinity within the CNMIOPs. Vibrating sample magnetometry (VSM) measurements using a Lake Shore VSM Model 7407 were conducted to analyze hysteresis loops, saturation magnetization (*Ms*), coercivity (*Hc*), and remanence (*Mr*) at room temperature. The stability of the samples was assessed using a Malvern Zetasizer Potential instrument (ZSP) by measuring zeta potential, which involved dispersion in distilled water, sonication, and subsequent zeta potential measurements at 25 °C. Data analysis was performed using Zetasizer Software 8.02 and the OriginLab Pro 2019 software. Additionally, the antioxidant activity of the CNMIOPs was evaluated using the DPPH test, with the inhibition of the DPPH free radical measured. The antioxidant potential was quantified as the *IC*_50_ value (mg/mL), representing the dose required to cause 50% inhibition, and calculated using the GraphPad Prism 9 software.

### 2.4. Preparation and Characterization of PCL/CNMIOP Membranes

To fabricate PCL/CNMIOPs membranes, solution electrospinning was employed following the methods outlined in a previous study [13]. Briefly, a 10% *w*/*v* PCL solution was prepared by dissolving PCL in a chloroform and DMF mixture with a volume ratio of 9:1 at room temperature, forming the electrospinning dope solution. CNMIOPs, prepared at pH = 7.5 and washed with ethanol or water, were dispersed in the dope solution at a concentration of 1.0 *w*/*w* using ultrasonication for 2 h. Electrospinning was performed using a laboratory-scale electrospinning machine (BioInicia, Fluidnatek LE-50 setup, Valencia, Spain) with the following parameters: 13 cm needle distance from a rotating drum collector, 0.9 mL/h feed rate, and 12 kV voltage.

Morphological and elemental information, composition, and nanoparticle distribution analyses within the electrospun PCL/CNMIOPs nanofibers were carried out using SEM and TEM techniques. Surface roughness profiling of the nanofiber surface was conducted using a Confocal Interferometric Optical Microscope (Sensofar S-NEOX, Sky Tech, Bukit Batok, Singapore) with an 8 µm amplitude and magnifications starting at 20X (ISO 4287), applying a 2CR-PC Filter [14]. Surface roughness was evaluated using two parameters: roughness average (*Ra*) and quadratic roughness (*Rq*). *Ra* represents the average absolute value of profile height deviations, while *Rq* is the root mean square average of the profile height deviation.

### 2.5. Statistical Analysis

The measurements were represented as the mean ± SD of three replicates to ensure accuracy in comparing the results. To determine significant differences between observations, a one-way ANOVA was performed. The level of significance (*p* < 0.05) was assessed using Duncan’s statistical analysis.

## 3. Results

### 3.1. FTIR

The FTIR spectra of the CNMIOPs prepared under different pH conditions (1.2, 7.5, and 12.5), followed by either ethanol or water washing (at pH 7.5), are presented in Figure 1.

Observations were performed in the range of 800–400 cm^−1^ (included in Figure 1), where magnetic Fe-O bands can be identified both as γ–Fe_2_O_3_/α–Fe_2_O_3_ and magnetite Fe_3_O_4_. The spectrum of magnetite (Fe_3_O_4_) shows distinct bands at 640–570 cm^−1^, along with shoulders at approximately 699 cm^−1^ and 448 cm^−1^, attributed to the Fe–O bond in the octahedral and tetrahedral positions, respectively. In contrast, maghemite exhibits several closely spaced bands within the range of 800–400 cm^−1^ [15]. The presence of bands at 586–571 cm^−1^ provides evidence that hematite α–Fe_2_O_3_ undergoes reduction, resulting in the formation of Fe_3_O_4_ [16]. Further analysis of the Fe_3_O_4_ FTIR spectra reveals new absorption bands around 1624 and 1390 cm^−1^, as well as peaks at 1284 and 1086 cm^−1^ [17]. However, the observed bonds at 739–641 cm^−1^ may be attributed to the maghemite phase (γ–Fe_2_O_3_), formed during the oxidation of magnetite during synthesis [18]. Additionally, the bands around 561, 542, and 461 cm^−1^ indicate the stretching vibration mode of Fe–O in the hematite phase (α–Fe_2_O_3_) [19]. The signal at 1133 cm^−1^ suggests the presence of the α–Fe_2_O_3_ phase, associated with crystalline Fe-O vibrations [20].

The observed bonds in the range of 3476–3421 cm^−1^ are attributed to the vibrational stretching of NaOH’s -OH groups [21]. Two distinct peaks between 2911 and 2833 cm^−1^ suggest hydrocarbon extension. The band at 1630 cm^−1^ corresponds to the deformation or stretching of aromatic rings or C=C vibration in alkane groups. The peak at approximately 1733 cm^−1^ is assigned to C=O bonds found in aldehydes, ketones, and esters. At 1380 cm^−1^, the band indicates the presence of ester groups [22]. After analyzing the spectra, we observed a reduction in FeCl_3_·6H_2_O with the oxygen atoms of -OH, which resulted in the splitting of the 1643 cm^−1^ band into three distinct peaks (1653, 1633, and 1623 cm^−1^). Interestingly, the spectra obtained from ethanol-washed CNMIOPs at pH 1.2 (Figure 1a) and water-washed CNMIOPs at pH 7.5 (Figure 1d) showed similar characteristics. However, ethanol-washed CNMIOPs at pH 7.5 and 12.5 exhibited more fragmentation, particularly at pH 7 (Figure 1b). This phenomenon provides valuable insights for quantifying the properties of NMIOPs, including oxide phase composition, crystal system type, and magnetic behavior, which will be further examined in the upcoming sections.

### 3.2. SEM

Figure 2 showcases the SEM images depicting the CNMIOPs prepared under different pH conditions (1.2, 7.5, and 12.5), followed by either ethanol or water washing (at pH 7.5).

The CNMIOPs washed with ethanol displayed polygonal structures with cubic and rhombohedral morphologies. In contrast, the CNMIOPs washed with water exhibited quasi-spherical structures with significant aggregation. Specifically, the ethanol-washed CNMIOPs prepared at pH 1.2 exhibited flower-like structures, multilayer formations, and some spherical aggregations. On the other hand, those prepared at pH values higher than seven displayed dispersible agglomeration with elongated and cubic morphologies. The water-washed CNMIOPs showcased spherical knots and aggregates. Several factors contributed to the observed morphologies, including interactions between nanoparticles due to their diverse polycrystalline structure morphology [23]. Additionally, the absence of a reducing agent with stabilizing properties can also have an impact on the observed morphologies [24].

For the ethanol washing system, the average diameters of the CNMIOPs ranged from 16.3 to 38.0 nm as the pH decreased, while for the water system at pH 7.5, an average diameter of 27.5 nm was observed. These results are summarized in Table 1 and will be further assessed using XRD and TEM analysis.

### 3.3. TEM

Figure 3 shows TEM images of CNMIOPs prepared under different pH conditions (1.2, 7.5, and 12.5), followed by either ethanol or water washing (at pH 7.5). Furthermore, Figure 3 also depicts the size distributions of the CNMIOPs.

As observed, ethanol-washed CNMIOPs at all pH levels displayed cubic and elongated particles, with increased elongation observed at pH 7.5 and 12.5, resulting in nanorods of lengths measuring 107.7 ± 61.5 nm and 77.1 ± 32.4 nm, respectively. In contrast, the water-washed CNMIOPs exhibited a diverse morphology comprising a mixture of spherical and fewer cubic structures. In addition, the ethanol-washed CNMIOPs exhibited average diameters ranging from 21.6 to 30.4 nm, showing a decreasing trend as the pH increased. On the other hand, the water-washed CNMIOPs (pH 7.5) displayed an average diameter of 27.1 nm, as indicated in Table 1.

This varied morphology strongly highlights the influence of both pH levels and washing solvents on the particle’s morphology, size, and shape. These findings make it an ideal choice for obtaining nanoparticles with customized properties.

### 3.4. XRD

Figure 4 displays the X-ray diffractograms of CNMIOPs, showcasing their crystallinity and phase composition (highlighted by red planes corresponding to magnetite and black planes corresponding to hematite), as influenced by varying pH values and washing solvents (ethanol or water).

The diffractograms clearly indicate that all the observed diffraction peaks can be attributed to the spinel structure [25]. Table 2 displays the 2θ (°) values and corresponding crystallographic reflection planes (hkl) of the various CNMIOPs.

It can be observed that the ratio of the different phases of CNMIOPs and their crystal systems (crystalline structures) are affected by the pH value and the washing solvent, as depicted in Figure 5 and Table 3.

The XRD quantification of the CNMIOPs revealed distinct crystalline structure anisotropy. Specifically, at low pH (pH = 1.2), the ethanol-washed CNMIOPs exhibited peaks corresponding to 97.4% polycrystalline hematite, aligned with the standard iron oxide powder diffraction patterns (JCPDS nº. 00-900-9782, 00-210-1167, and 00-210-8027), and 2.6% polycrystalline magnetite, aligned with the standard iron oxide powder diffraction patterns (JCPDS nº. 00-153-2800 and 00-152-6955) [26,27,28,29]. Upon increasing the pH to neutral values (7.5) and washing with ethanol, the XRD peaks corresponded to 28.5% polycrystalline hematite (JCPDS nº. 00-900-5841 and 00-900-9782) and 71.5% cubic magnetite (JCPDS nº.00-230-0616 and 00-230-0617) [30,31]. Nevertheless, when the CNMIOPs prepared at pH 7.5 were washed with water, the XRD peaks once again indicated 82.6% polycrystalline hematite (JCPDS nº. 00-210-8027, 00-152-6955, and 00-210-1167) and 17.4% polycrystalline magnetite (JCPDS nº.00-101-1240, and 00-153-2800) [26,27,28,29]. Upon further increasing the pH value to 12.5 (ethanol-washed CNMIOPs), the XRD peaks corresponded to 13.5% trigonal hematite (hexagonal axis) (JCPDS nº. 00-722-8110) and an 86.5% crystalline cubic structure of magnetite (JCPDS nº. 00-900-0139) [32,33].

The sizes of the CNMIOPs were calculated using the Debye–Scherrer equation, as mentioned earlier. Table 3 provides information about the sizes of the nanoparticles and their crystallinity proportions obtained from XRD analysis.

At pH = 1.2, the ethanol-washed CNMIOPs exhibited a polycrystalline structure with an average diameter of 39.5 nm. As the pH value increased to 7.5 (with ethanol washing), the average size of the nanoparticles decreased to 26.6 nm, accompanied by an increase in the proportion of Fe_3_O_4_. Further increasing the pH to 12.5 (with ethanol washing) resulted in a continued decrease in particle size to 16.8 nm (Table 3), while the Fe_3_O_4_ proportion increased. When considering the effect of the washing solvent at pH 7.5, the water-washed CNMIOPs showed a larger size (29.6 nm, Table 3) compared with those washed with ethanol.

The XRD analysis revealed diverse crystal systems with varying degrees of crystallinity, particularly noticeable at higher pH levels. The pH influenced particle size and the hydrolysis of Fe^3+^/Fe^2+^ ions, resulting in a decrease in CNMIOP size as pH increased [34]. This can be attributed to fast hydrolysis rates and phase transitions. H^+^ ions played a dual role in slowing down Fe^3+^ hydrolysis and stabilizing oxygen-terminated crystal planes, such as α–Fe_2_O_3_ [8]. Optimal conditions for nearly pure magnetite formation were observed at pH > 7.5–12.5, with minimal associated iron oxide. Lower pH values favored the formation of acicular Fe_2_O_3_-NPs. An increase in pH led to well-defined cubic particles, with limited reversion to α–Fe_2_O_3_/γ–Fe_2_O_3_ due to the larger reaction volume. This limitation may suggest the absence of maghemite (γ–Fe_2_O_3_) peaks in the XRD analysis. Furthermore, the absence of maghemite peaks in the XRD analysis could be attributed to either a lower concentration or the presence of an undetectable amorphous form. It is important to note that all samples exhibited crystallinity higher than 80%.

### 3.5. Magnetic Properties

Figure 6 shows the magnetization (*M*) vs. magnetic field (*H*) hysteresis curves of the CNMIOPs prepared under different pH conditions (1.2, 7.5, and 12.5), followed by either ethanol or water washing (at pH 7.5). The magnetic properties of the CNMIOPs are summarized in Table 4.

Table 4 presents a clear distinction between all the systems in terms of *Ms* values, with pH 12.5 combined with ethanol yielding the highest value. Notably, the magnetic properties of CNMIOPs revealed substantial differences in coercivity (*Hc*) values across all pH levels and washing solvents. The utmost *Hc* value was achieved at pH 7.5 with water as the washing solvent. Moreover, the remanence (*Mr*) values exhibited significant variations across different pH levels and washing solvents, with the highest *Mr* value observed at pH 7.5 when ethanol was employed for washing. It is worth mentioning that the *Mr/Ms* ratio displayed marked differences between all pH levels and washing solvents, reaching its peak at pH 1.2 with ethanol, as pH and the washing process significantly influenced the magnetic properties of the nanoparticles. The XRD analysis revealed phase changes, including the formation of magnetite and hematite mixtures, with various crystal systems. Despite the expected superparamagnetism (lower than 20 nm) at room temperature, interactions between the particles led to coercivity, even with weak dipolar interactions in the solvent [35]. Interestingly, the observed fragmentations in the FTIR spectra correlate with the magnetic properties, as shown in Figure 1a–c. The CNMIOPs with the lowest magnetic saturation exhibited smooth absorbance without fragmentations (Figure 1d).

The magnetic properties of the CNMIOPs were influenced significantly by pH and the washing process. At pH 1.2 (washed with ethanol), the CNMIOPs mainly consisted of hematite (Fe_2_O_3_, 97.4%), with an *Ms* of 5.0 emu/g. This *Ms* value could be attributed to the presence of a higher proportion of monoclinic hematite (57.5%). With increasing hydroxyl ions (OH^-^), the CNMIOPs at pH 7.5 (with 71.5% cubic Fe_3_O_4_) exhibited mixed phases of magnetite and hematite, resulting in an *Ms* of 57.5 emu/g. At pH 12.5 (with 86.5% cubic Fe_3_O_4_), smaller magnetite nanoparticles formed, showing ferromagnetic properties with an *Ms* of 58.3 emu/g.

The increases in the cubic structure, magnetite phase, and smaller nanoparticle size contributed to improved magnetic response. The presence of trigonal structures in both iron oxide phases hindered magnetic properties, favoring inter-crystalline interactions. Similarly, the higher proportion of monoclinic hematite at pH 1.2 (washed with ethanol) showed increased saturation magnetization compared with the CNMIOPs at pH 7.5 (washed with water), which can be attributed to different orientations of the polycrystalline microstructure. In contrast, the CNMIOPs at pH 7.5 and 12.5 displayed smaller cubic magnetite nanoparticles and elongated nanoparticles (Figure 2b,c), resulting in improved saturation magnetization.

Overall, pH-dependent crystal structures and nanoparticle sizes played a crucial role in the magnetic behavior of the CNMIOPs. Previous studies have shown that elongated iron oxide nanoparticles exhibit distinct magnetic properties compared with nearly spherical ones, primarily because of the increased importance of the shape anisotropy term within the total magnetic anisotropy [36]. The observed reduction in the saturation magnetization values of the CNMIOPs upon water washing can be attributed to their propensity to agglomerate or form aggregates in an aqueous medium [37]. Furthermore, the CNMIOPs washed with water exhibited spherical morphology and displayed the highest coercivity value of 268 Oe. In contrast, the CNMIOPs washed with ethanol showed lower coercivity, which may be attributed to a reduction in local magneto-crystalline anisotropy caused by defects at the boundaries [38]. Interestingly, the TEM analysis revealed that the CNMIOPs (pH ≥ 7.5, washed with ethanol) exhibited an elongated rod-like shape, possibly facilitated by the presence of ethanol molecules that promoted the connection of the central parts of the quasi-spherical nanoparticles, resulting in unidirectional growth [36]

These results underscore the potential for tailoring the magnetic behavior of nanoparticles by manipulating their pH values and the washing process. Additionally, the disparities observed in these values between the two types of nanoparticles emphasize the significant impact of composition and processing on the magnetic properties of the particles.

### 3.6. Nanoparticle Stability

Figure 7 illustrates the assessment of CNMIOP stability using zeta potential measurements (ζ value, mV) conducted in distilled water.

Table 4 provides a comprehensive summary of the zeta potential measurements conducted on the CNMIOPs. Notably, these measurements demonstrate significant variations between different pH values and washing solvents. It is worth mentioning that all samples exhibited negative ζ values, indicating a range of particle stability from stable to less stable. Among the different conditions, the water-washed CNMIOPs prepared at pH 7.5 displayed the highest negative stability value with a ζ value of −23.8 mV. In contrast, the ethanol-washed CNMIOPs prepared at the same pH exhibited a lower ζ value of −17.0 mV. Interestingly, the ethanol-washed CNMIOPs demonstrated the lowest ζ value of −14.7 mV. Moreover, it is of particular interest to explore the relationship between these measurements and other factors, such as nanoparticle size, shape, crystallinity, oxide phase, and aggregation. The use of ethanol as a washing system resulted in a smaller CNMIOP size, which subsequently led to a decrease in stability, as depicted in Figure 7. This may be due to their higher surface-area-to-volume ratio, which increases surface energy and susceptibility to agglomeration [39]. Previous studies have reported similar zeta potential values for conventional silver and iron nanoparticles [39,40]. The stability of irregularly shaped CNMIOPs was slightly influenced by changes in crystallinity and oxide phase, without a clear relationship observed. Notably, the CNMIOPs at pH 12.5 (washed with ethanol), with high crystallinity (97%), showed the lowest stability (a ζ value of −14.7 mV), potentially because of the absence of reducing agents [24]. The zeta potential measurements exhibited a strong correlation with the magnetic properties of the nanoparticles. Notably, the CNMIOPs at pH 12.5 (ethanol-washed) displayed the highest magnetization saturation (58.3 emu/g), which coincided with the lowest stability (ζ = −14.7 mV). Conversely, the CNMIOPs at pH 7.5 (water-washed) with the lowest magnetization saturation (1.7 emu/g) demonstrated the highest stability (ζ = −23.8 mV). These findings suggest that magnetic interactions between particles facilitated clustering and aggregation within the solution [41]. The stability of CNMIOPs prepared with different pH levels and washing processes is crucial for ensuring their robustness and suitability across various applications. The findings presented in this study demonstrate the ability of CNMIOPs to maintain their structural integrity and functionality in neutral environments. Nevertheless, further studies are needed to explore the stability of CNMIOPs across a broader range of pH levels, including acidic and alkaline conditions. This stability is particularly significant in applications where nanoparticles may encounter diverse pH environments, such as drug delivery systems or biomedical applications. Our results suggest that CNMIOPs exhibit the capacity to maintain their stability and performance in water, ensuring their reliability and effectiveness in targeted applications. Furthermore, their performance can be enhanced through an extensive exploration of the parameters and conditions of the synthesis.

### 3.7. Antioxidant Activity

The functional properties of CNMIOPs were evaluated for their antioxidant activity against DPPH. Figure 8 illustrates the antioxidant activity expressed by the *IC*_50_ value of inhibition for the prepared nanoparticles, highlighting their pH-dependent behavior in correlation with the chosen preparation method (Table 1).

The *IC*_50_ values for the CNMIOPs were found to be 12.1, 4.8, and 2.1 mg/mL at pH values of 1.2, 7.5, and 12.5, respectively, when washed with the ethanol, and they were found to be 7.1 mg/mL at pH 7.5 when washed with H_2_O. Notably, the *IC*_50_ values for the CNMIOPs were found to vary significantly depending on the pH and washing solvent used. The ethanol-washed CNMIOPs at pH 12.5 demonstrated the most significant *IC*_50_ value. This can be attributed to a combination of factors, including a high proportion of Fe_3_O_4_, a smaller particle size, higher crystallinity, and increased magnetic properties (*Ms*). These findings are consistent with similar trends reported in previous investigations [42,43]. The antioxidant activity of CNMIOPs prepared with different pH levels and washing solvents is an essential attribute that enhances their potential applications in various fields, particularly in biomedical and therapeutic settings. Antioxidants play a vital role in reducing oxidative stress and neutralizing harmful free radicals, which are associated with various diseases and cellular damage. However, it should be noted that the ability of CNMIOPs to maintain their antioxidant activity across different conditions is limited. Nonetheless, their potential for combating oxidative stress in diverse physiological environments can be enhanced through various settings, including those explored in this study. This finding suggests that there is room for further improvement and utilization of CNMIOPs in formulations targeting oxidative-stress-related diseases or in systems where antioxidant properties are desired, such as tissue-engineering scaffolds.

### 3.8. SEM Imaging of Nanofibers

The application of CNMIOPs to electrospun membranes was investigated, focusing on samples with higher or lower magnetic properties. The effect of incorporating these CNMIOPs, prepared at pH 7.5 and washed with either ethanol or water, on the morphology of the electrospun nanofibers was studied. Figure 9 presents the morphologies of the electrospun nanofiber membranes for both PCL and PCL/CNMIOP samples.

The nanofibers of the PCL membranes in the absence of CNMIOPs displayed a consistent and even morphology. However, when CNMIOPs were incorporated, irregularities appeared on the nanofiber surface. This suggests that the properties of CNMIOPs, such as their tendency to agglomerate, granulate, and encapsulate within fibers, play a role during the electrospinning process. Both ethanol and water-washed CNMIOPs resulted in broader fiber zones compared with the PCL membranes without CNMIOPs. Specifically, the CNMIOPs prepared at pH 7.5 displayed larger aggregation knots, leading to thicker or more stretched fibers. On the other hand, the CNMIOPs washed with ethanol showed more homogeneous aggregation knot length and shape. This behavior can be attributed to various factors, such as particle size, crystallinity, and shape. Previous studies have also observed similar trends, where systems with larger spherical nanoparticles tend to form larger aggregates, while elongated nanoparticles contribute to a more dispersed distribution of broadened fibers [44]. Figure 10 illustrates the fiber diameter distribution for both PCL membranes without CNMIOPs and those with incorporated ethanol- or water-washed CNMIOPs.

Incorporating CNMIOPs prepared at pH 7.5 and washed with either ethanol or water led to variations in the average diameter of the nanofibers, highlighting the unique characteristics of these nanoparticles (Table 5).

### 3.9. Surface Roughness of Nanofibers

The effect of incorporating the CNMIOPs prepared at pH 7.5 and washed with either ethanol or water on the roughness surface of the electrospun nanofiber surface was studied. Figure 11 presents 2D surface plots of the electrospun nanofiber membranes for both the PCL and PCL/CNMIOP samples.

Table 5 presents the impact of incorporating CNMIOPs on the roughness of the nanofiber surface, as indicated by the *Ra* and *Rq* values. It was observed that the inclusion of CNMIOPs, regardless of whether they were washed with ethanol or water, led to higher *Ra* and *Rq* values compared with the PCL electrospun membrane alone. The highest roughness was observed in the case of CNMIOPs prepared at pH 7.5 and washed with water. This can be attributed to the presence of larger aggregation knots within the fibers, which arise from the larger size of CNMIOPs prepared under pH 7.5 and washed with water. Additionally, the interaction between the different crystalline systems of the nanoparticles contributed to the formation of larger knot aggregations. These findings align with the results obtained from SEM analyses and are in line with previous studies, providing further support for the observed effects [45,46]. The roughness of membrane surfaces holds paramount importance in the growth and adherence of diverse cell lines, making it a vital parameter with significant implications for biomaterial applications. Its impact on cellular interactions highlights the critical role of surface roughness in determining the compatibility and effectiveness of materials.

### 3.10. TEM Imaging of Nanofibers

Figure 12 showcases the TEM images, revealing the morphologies of electrospun nanofiber membranes consisting of PCL without CNMIOPs, as well as those with incorporated ethanol- or water-washed CNMIOPs prepared at pH 7.5.

CNMIOPs prepared at pH 7.5 and washed with ethanol demonstrate superior dispersion and a more uniform distribution, exhibiting minimal agglomeration. In contrast, CNMIOPs prepared at pH 7.5 and washed with water exhibit higher levels of aggregation, potentially because of the increased hydrophobic nature acquired during the washing process. Another factor contributing to aggregation could be the interactions between the charged surfaces of CNMIOPs and the PCL chains within the fibers, resulting in structural changes [47]. This phenomenon is noteworthy in terms of surface roughness, as both types of CNMIOPs contribute to modifications in the fiber surface. The enhanced properties of CNMIOPs, including their magnetic properties, antioxidant activity, good dispersity, and ability to increase surface roughness, contribute to their potential applications in various fields.

One potential application of PCL electrospun materials infused with CNMIOPs is tissue engineering. The magnetic properties of CNMIOPs can enable the remote manipulation and guidance of the scaffold within the body using external magnetic fields. This feature holds promise for applications such as targeted drug delivery and tissue regeneration. The incorporation of CNMIOPs into an electrospun PCL matrix can enhance the overall mechanical properties of the scaffold, improving its structural integrity and providing a suitable microenvironment for cell attachment, proliferation, and differentiation [13].

## 4. Conclusions

The pH value and washing solvent have a significant impact on the magnetic properties of chemical nanomagnetic iron oxide particles (CNMIOPs). Lower pH values and water washing promote particle aggregation, affecting the magnetic behavior of CNMIOPs. Further investigation is needed to understand the variations in response under different conditions. However, by carefully selecting processing conditions such as pH values and solvents, it is possible to control and enhance the magnetism of CNMIOPs. Chemical synthesis at higher pH values (7.5 and 12.5) and ethanol washing result in improved magnetic properties and antioxidant properties. This study highlights the importance of considering pH and washing solvent when synthesizing magnetic nanoparticles and emphasizes the need for detailed characterization to ensure reproducibility and maximize the potential of CNMIOPs in various applications. Additionally, this study demonstrates the influence of pH and washing solvent on the morphology, and roughness of nanofiber membranes containing CNMIOPs, emphasizing the role of CNMIOP–PCL interactions. In conclusion, the findings of this study provide valuable insights for customizing the chemical synthesis of CNMIOPs, positioning them as potentially valuable additives in the pharmaceutical and food industries. These magnetic nanoparticles can serve as effective drug carriers in complex systems, offering a promising solution for various applications.

CNMIOPs prepared in this study displayed enhanced, unique properties, including magnetic response, antioxidant activity, good dispersity, and surface roughness enhancement, making them highly versatile for various applications. In the context of tissue engineering, the incorporation of CNMIOPs into PCL electrospun materials offers promising potential. The magnetic properties enable remote manipulation and targeted drug delivery while improving the scaffold’s mechanical integrity. These advancements create an optimal microenvironment for cell attachment, proliferation, and differentiation. Thus, the combination of CNMIOPs with PCL electrospun materials holds great promise for tissue-engineering applications.

However, it is important to acknowledge potential limitations in this study, including the specific conditions and characterization techniques used, which may limit the generalizability of the findings. Additionally, further exploration of factors influencing the properties of the nanoparticles is needed. To enhance our understanding, future research should focus on examining the biocompatibility, toxicity, and long-term storage effects of CNMIOPs, particularly for biomedical applications. Investigating the influence of factors such as temperature, reaction time, and reactant concentration on the properties of CNMIOPs is crucial for a comprehensive understanding. Moreover, it would be highly beneficial to explore green methods or combine different approaches to enhance the diverse, unique properties of CNMIOPs and scale up their synthesis for industrial applications. By considering these limitations and focusing on these research areas, we can advance our understanding of CNMIOPs and unlock their full potential for various applications.

## Figures and Tables

**Figure 1 nanomaterials-13-02242-f001:**
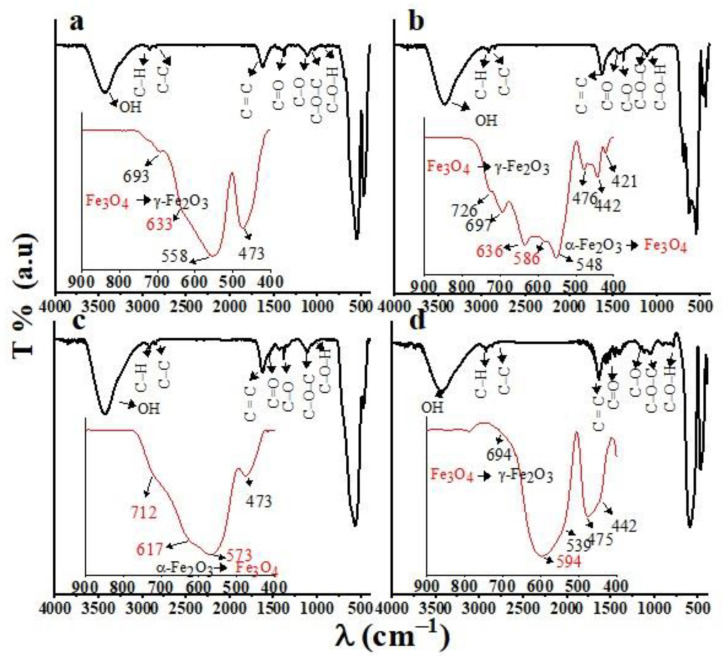
FTIR spectra of CNMIOPs synthesized under various pH conditions: (**a**) pH = 1.2; (**b**) pH = 7.5; (**c**) pH = 12.5, with subsequent ethanol washing; and (**d**) pH = 7.5 followed by water washing. Colored lines represent the FTIR spectra in the fingerprint region 800–400 cm^−1^, while red font indicates the magnetite phase, and black font indicates the hematite phase.

**Figure 2 nanomaterials-13-02242-f002:**
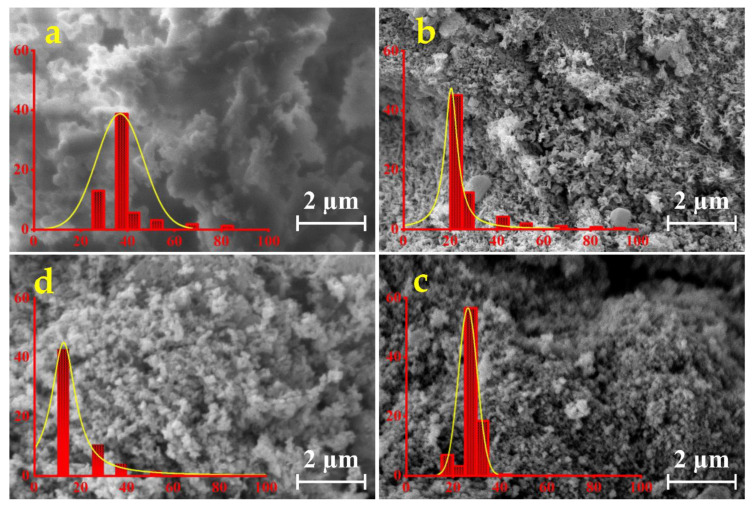
SEM images and their corresponding diameter distribution histograms (in red color, represented in nanometers) of CNMIOPs synthesized under various pH conditions: (**a**) pH = 1.2; (**b**) pH = 7.5; (**c**) pH = 12.5, with subsequent ethanol washing; and (**d**) pH = 7.5 followed by water washing.

**Figure 3 nanomaterials-13-02242-f003:**
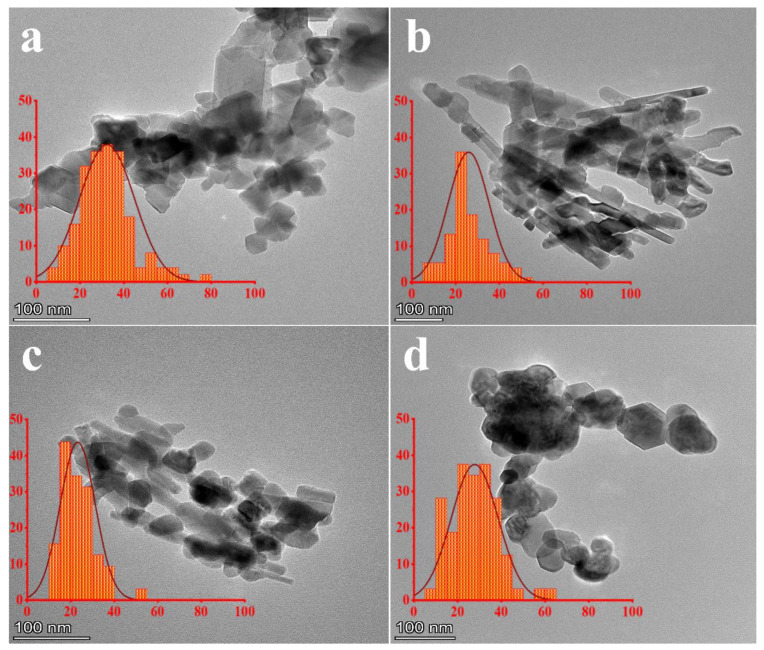
TEM images and their corresponding diameter distribution histograms (in red color, represented in nanometers) of CNMIOPs synthesized under various pH conditions: (**a**) pH = 1.2; (**b**) pH = 7.5; (**c**) pH = 12.5, with subsequent ethanol washing; and (**d**) pH = 7.5 followed by water washing.

**Figure 4 nanomaterials-13-02242-f004:**
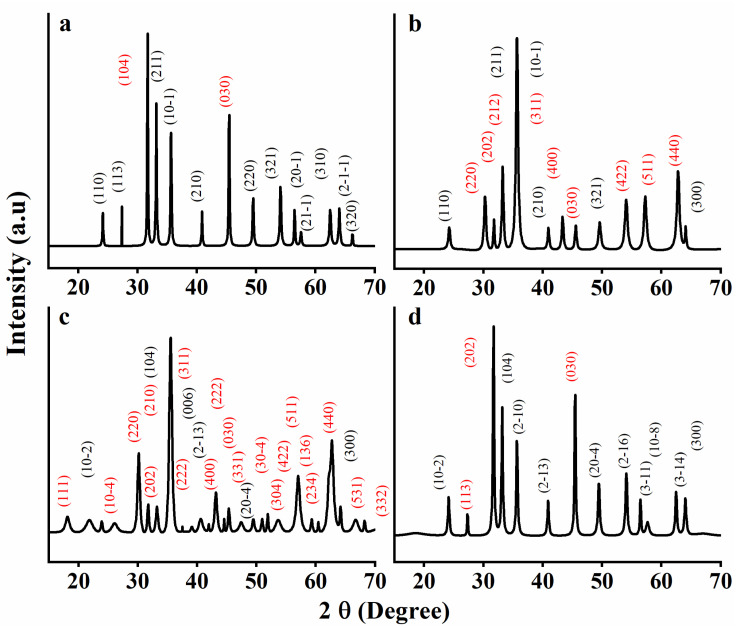
XRD spectra of CNMIOPs synthesized under various pH conditions: (**a**) pH = 1.2; (**b**) pH = 7.5; (**c**) pH = 12.5, with subsequent ethanol washing; and (**d**) pH = 7.5 followed by water washing. Peaks highlighted by red planes corresponding to magnetite and black planes corresponding to hematite.

**Figure 5 nanomaterials-13-02242-f005:**
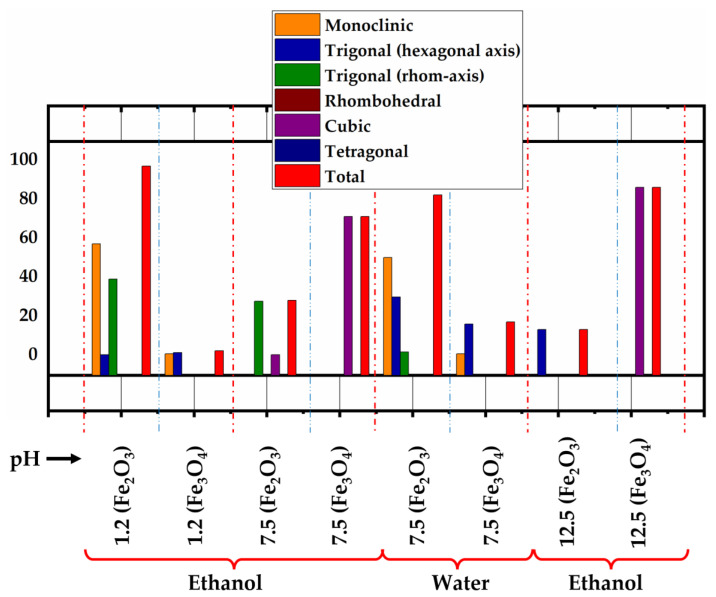
The different crystal systems attributed to the Fe_2_O_3_ and Fe_3_O_4_ proportions of CNMIOPs synthesized under various pH conditions with subsequent ethanol washing (1.2, 7.5, 12.5) or water washing (at pH = 7.5). Red dotted lines indicate the starting and final conditions, while blue dotted lines indicate phase separation within the same condition.

**Figure 6 nanomaterials-13-02242-f006:**
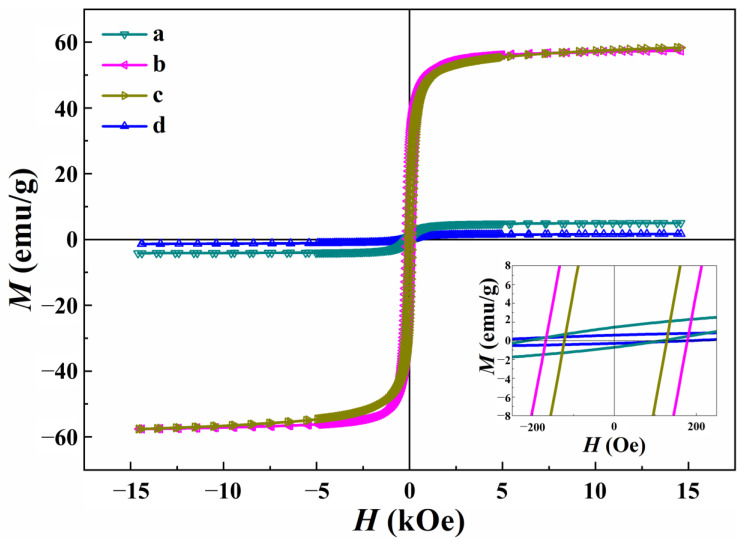
Magnetic hysteresis curves at room temperature of CNMIOPs synthesized under various pH conditions: (**a**) pH = 1.2; (**b**) pH = 7.5; (**c**) pH = 12.5, with subsequent ethanol washing; and (**d**) pH = 7.5 followed by water washing.

**Figure 7 nanomaterials-13-02242-f007:**
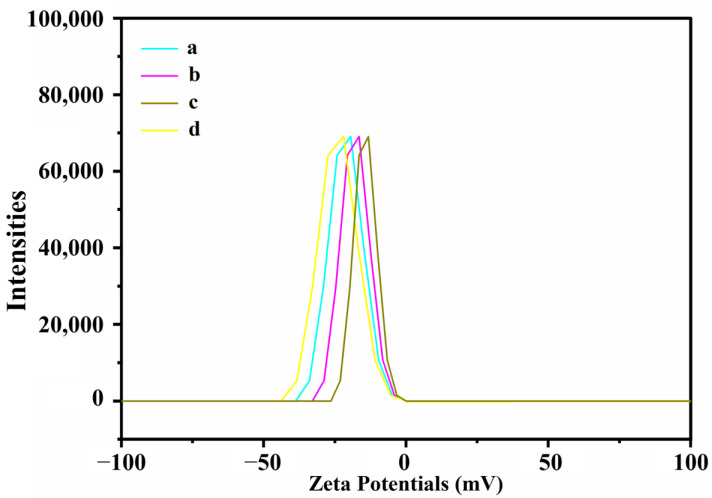
Zeta potential values **(**ζ value, mV) of CNMIOPs synthesized under various pH conditions: (**a**) pH = 1.2; (**b**) pH = 7.5; (**c**) pH = 12.5, with subsequent ethanol washing; and (**d**) pH = 7.5 followed by water washing.

**Figure 8 nanomaterials-13-02242-f008:**
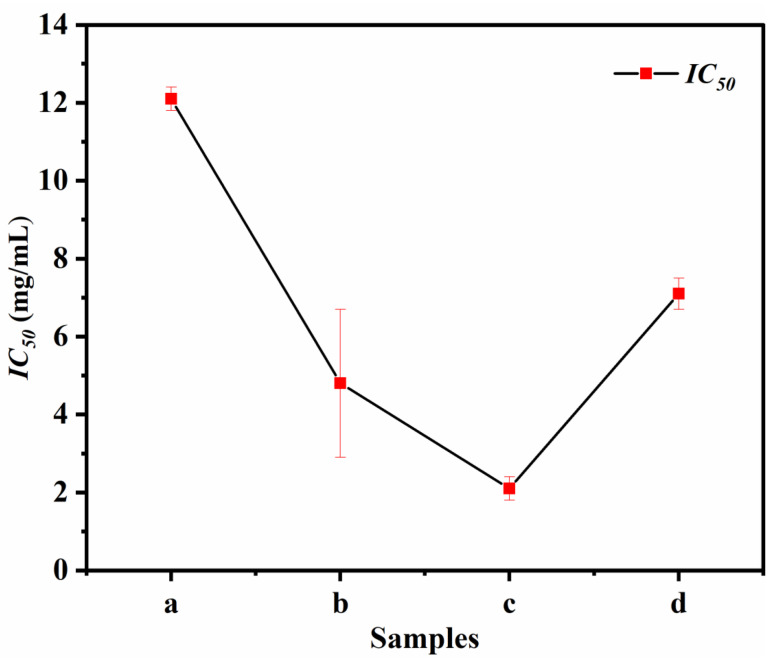
*IC*_50_ value in mg/mL of inhibition for CNMIOPs synthesized under various pH conditions: (**a**) pH = 1.2; (**b**) pH = 7.5; (**c**) pH = 12.5, with subsequent ethanol washing; and (**d**) pH = 7.5 followed by water washing.

**Figure 9 nanomaterials-13-02242-f009:**
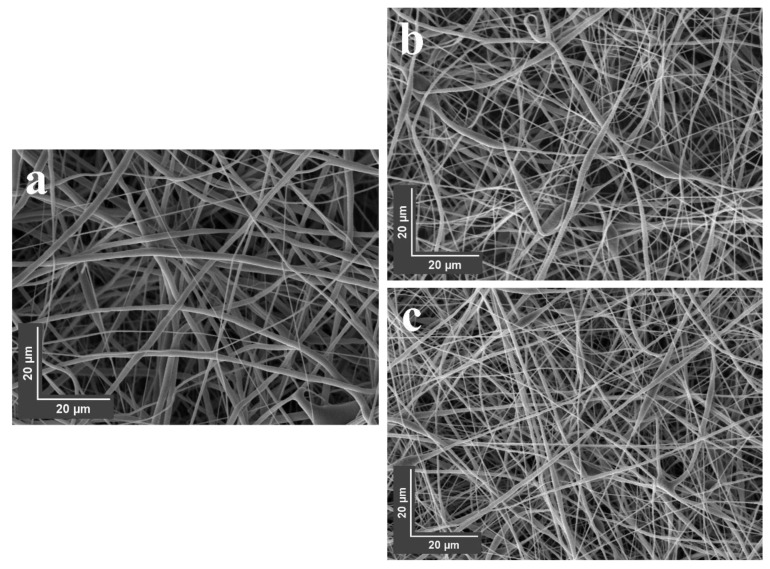
SEM images of electrospun nanofibers obtained from (**a**) pure PCL, (**b**) PCL/CNMIOPs prepared at pH 7.5 and washed with ethanol, and (**c**) PCL/CNMIOPs prepared at pH = 7.5 and washed with water.

**Figure 10 nanomaterials-13-02242-f010:**
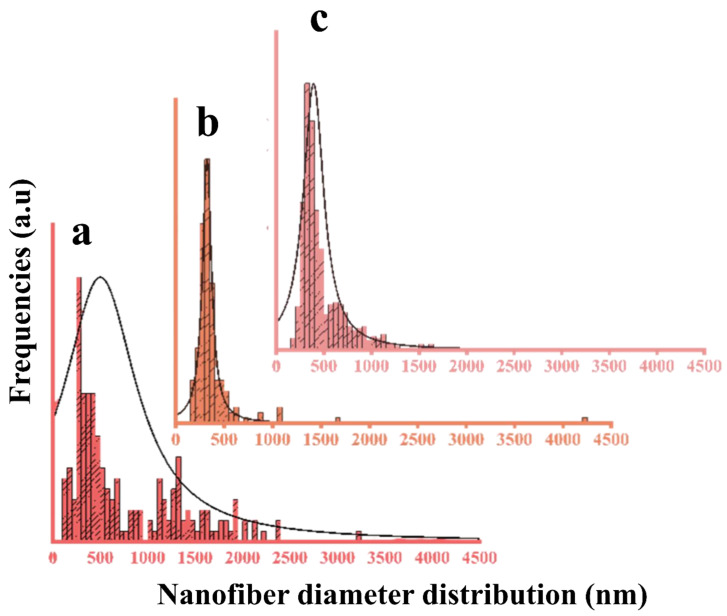
Diameter distributions of electrospun nanofiber membranes obtained from (**a**) pure PCL, (**b**) PCL/CNMIOPs prepared at pH 7.5 and washed with ethanol, and (**c**) PCL/CNMIOPs prepared at pH = 7.5 and washed with water.

**Figure 11 nanomaterials-13-02242-f011:**
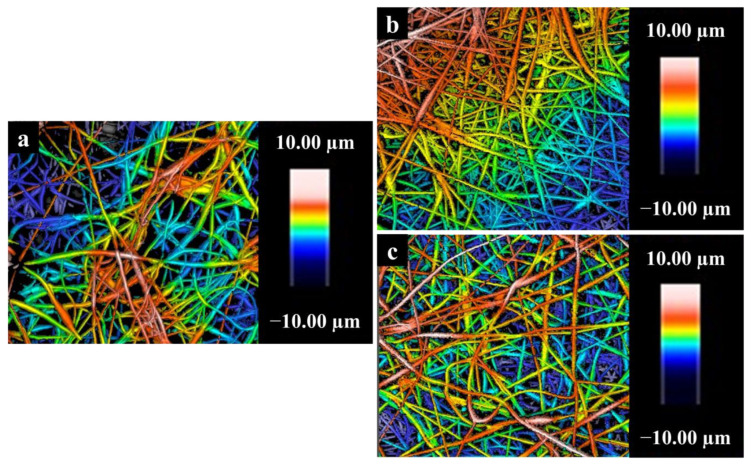
Two-dimensional surface plot of electrospun nanofibers obtained from (**a**) pure PCL, (**b**) PCL/CNMIOPs prepared at pH 7.5 and washed with ethanol, and (**c**) PCL/CNMIOPs prepared at pH = 7.5 and washed with water.

**Figure 12 nanomaterials-13-02242-f012:**
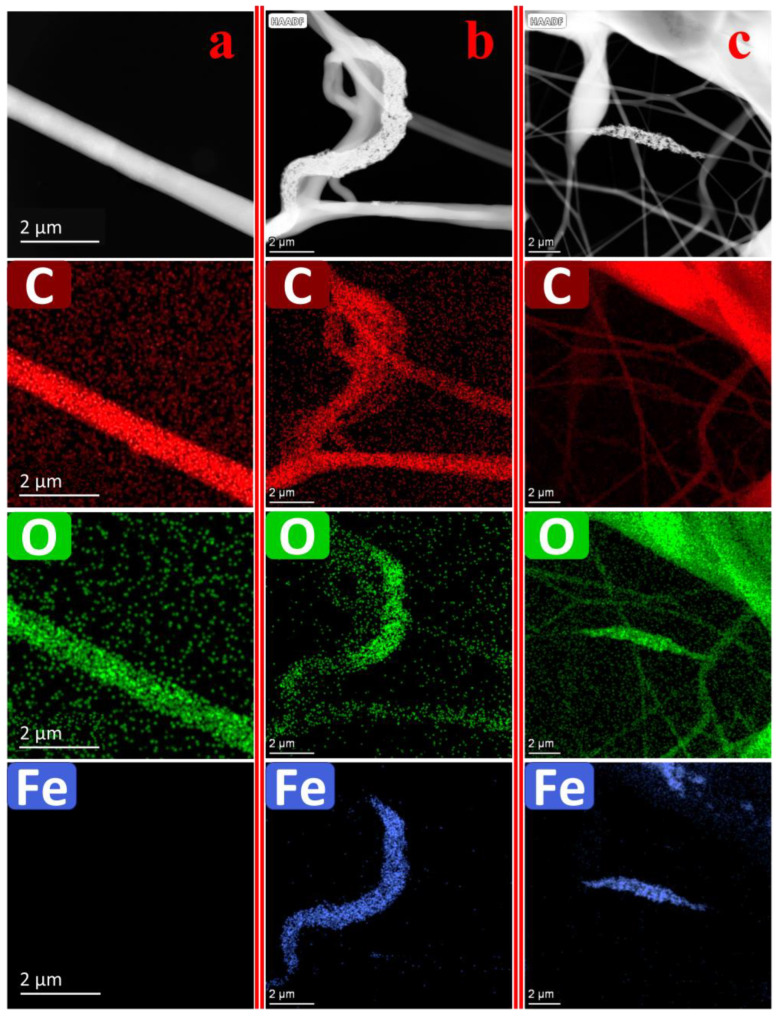
TEM images of electrospun nanofiber membranes obtained from (**a**) pure PCL, (**b**) PCL/CNMIOPs prepared at pH 7.5 and washed with ethanol, and (**c**) PCL/CNMIOPs prepared at pH = 7.5 and washed with water.

**Table 1 nanomaterials-13-02242-t001:** Characterization and antioxidant activity of the CNMIOPs prepared under different pH conditions (1.2, 7.5, and 12.5), followed by either ethanol or water washing (at pH 7.5): average size (nm) from SEM threshold, TEM images, and XRD measurements; crystallinity degree; and *IC*_50_ DPPH free radical scavenging activity (mg/mL).

Sample	CNMIOPs			
Conditions	1.2	7.5		12.5
		Ethanol	Water	
D _SEM_ (nm)	38.0 ± 1.9 ^a^	25.4 ± 0.6 ^c^	27.5 ± 0.5 ^b^	16.3 ± 4.6 ^d^
D _TEM_ (nm)	30.4 ± 0.5 ^a^	24.4 ± 0.5 ^c^	27.1 ± 0.9 ^b^	21.6 ± 0.3 ^d^
D _XRD_ (nm)	39.5 ± 5.0 ^a^	26.6 ± 3.3 ^c^	29.6 ± 0.9 ^b^	16.8 ± 1.4 ^d^
Crystallinity %	84.8	93.0	85.6	97.0
*IC*_50_ (mg/mL)	12.1 ± 0.3 ^a^	4.8 ± 1.9 ^c^	7.1 ± 0.4 ^b^	2.1 ± 0.3 ^d^

Different superscript letters (a–d) within a row indicate significant differences between mean observations (*p* < 0.05).

**Table 2 nanomaterials-13-02242-t002:** Crystallographic reflection planes and 2θ (°) values of CNMIOPs prepared under different pH conditions (1.2, 7.5, and 12.5), followed by either ethanol or water washing (at pH 7.5): Fe^+3^ indicates hematite phase (Fe_2_O_3_), and Fe^+2^ indicates magnetite phase (Fe_3_O_4_).

CNMIOPs							
1.2		7.5				12.5	
		Ethanol		Water			
2θ (°)	planes	2θ (°)	planes	2θ (°)	planes	2θ (°)	planes
24.18	(110) Fe^+3^	24.24	(110) Fe^+3^	18.30	(003) Fe^+3^	18.21	(111) Fe^+2^
27.36	(113) Fe^+3^	30.30	(220) Fe^+2^	24.12	(10-2) Fe^+3^	21.85	(10-2) Fe^+3^
31.70	(104) Fe^+2^	31.77	(202) Fe^+2^	26.58	(113) Fe^+2^	23.93	(10-4) Fe^+2^
33.17	(211) Fe^+3^	33.20	(211) Fe^+3^	31.70	(202) Fe^+2^	26.11	(220) Fe^+2^
35.68	(10-1) Fe^+3^	33.23	(212) Fe^+2^	33.17	(104) Fe^+3^	30.17	(202) Fe^+2^
40.88	(210) Fe^+3^	35.66	(311) Fe^+2^	35.62	(2-10) Fe^+3^	31.76	(210) Fe^+2^
45.47	(030) Fe^+2^	35.69	(10-1) Fe^+3^	40.88	(2-13) Fe^+3^	33.23	(104) Fe^+3^
49.51	(220) Fe^+3^	40.94	(210) Fe^+3^	45.47	(030) Fe^+2^	35.56	(311) Fe^+2^
54.09	(321) Fe^+3^	43.34	(400) Fe^+2^	49.45	(20-4) Fe^+3^	37.51	(222) Fe^+2^
56.48	(20-1) Fe^+3^	45.59	(030) Fe^+2^	54.09	(2-16) Fe^+3^	39.14	(006) Fe^+3^
57.57	(21-1) Fe^+3^	49.60	(321) Fe^+3^	56.48	(3-11) Fe^+3^	40.62	(2-13) Fe^+3^
62.48	(310) Fe^+3^	54.05	(422) Fe^+2^	57.66	(10-8) Fe^+3^	41.96	(400) Fe^+2^
64.02	(2-1-1) Fe^+3^	57.27	(511) Fe^+2^	62.48	(3-14) Fe^+3^	43.14	(222) Fe^+2^
66.21	(320) Fe^+3^	62.78	(440) Fe^+2^	64.01	(300) Fe^+3^	44.55	(030) Fe^+2^
		64.08	(300) Fe^+3^			45.35	(311) Fe^+2^
						47.43	(20-4) Fe^+3^
						49.44	(30-4) Fe^+2^
						50.97	(304) Fe^+2^
						51.89	(422) Fe^+2^
						53.67	(511) Fe^+2^
						57.03	(136) Fe^+2^
						60.44	(234) Fe^+2^
						62.72	(440) Fe^+2^
						64.19	(300) Fe^+3^
						66.73	(531) Fe^+2^
						68.23	(332) Fe^+2^

**Table 3 nanomaterials-13-02242-t003:** Crystal systems and proportions of Fe_2_O_3_ and Fe_3_O_4_ proportions in CNMIOPs prepared under different pH conditions (1.2, 7.5, and 12.5), followed by either ethanol or water washing (at pH 7.5). Fe^+3^ indicates total hematite (Fe_2_O_3_), and Fe^+2^ indicates total magnetite (Fe_3_O_4_).

Samples			CNMIOPs					
Conditions	1.2		7.5				12.5	
			Ethanol		Water			
Crystal system	Fe^+3^ (%)	Fe^+2^ (%)	Fe^+3^ (%)	Fe^+2^ (%)	Fe^+3^ (%)	Fe^+2^ (%)	Fe^+3^ (%)	Fe^+2^ (%)
Monoclinic	57.5	1			50.5	1.1		
Space Group	C 1 2/c 1	P 1 2/c 1			C 1 2/c 1	P 1 2/c 1		
Trigonal (Hexagonal Axis)	0.6	1.6			30.2	16.3	13.5	
Space Group	R-3 c:H	R-3 m:H			R-3 c:H	R-3 m:H	R-3 c:H	
Trigonal (Rhombohedral Axis)	39.3		28		1.9			
Space Group	R-3 c:R		R-3 c:R		R-3 c:R			
Rhombohedral								
Space Group								
Cubic			0.5	71.5				86.5
Space Group			P 43 3 2	F d-3 m:2				F d 3 m:1
Tetragonal								
Total	97.4	2.6	28.5	71.5	82.6	17.4	13.5	86.5

**Table 4 nanomaterials-13-02242-t004:** Results obtained for the magnetic properties of CNMIOPs prepared under different pH conditions (1.2, 7.5, and 12.5), followed by either ethanol or water washing (at pH 7.5): saturation magnetization (*Ms*, emu/g); coercivity (*H_c_*, Oe); magnetic remanence (*Mr*, emu/g); and zeta potential (ζ, mV). Different superscript letters (a–d) within a row indicate significant differences between mean observations (*p* < 0.05).

Sample	CNMIOPs			
Conditions	1.2	7.5		12.5
		Ethanol	Water	
*Ms* (emu/g)	5.0 ^c^	57.5 ^b^	1.7 ^d^	58.3 ^a^
*Hc* (Oe)	163.1 ^c^	173.4 ^b^	268 ^a^	125.8 ^d^
*Mr* (emu/g)	1.2 ^c^	28.8 ^a^	0.5 ^d^	23.0 ^b^
*Mr*/*Ms*	0.20	0.50	0.30	0.39
ζ (mV)	−21.0 ^b^	−17.0 ^c^	−23.8 ^a^	−14.7 ^d^

**Table 5 nanomaterials-13-02242-t005:** Nanofiber diameter (nm), roughness (*Ra,* nm), and quadratic root mean square average roughness (*Rq,* nm) for electrospun nanofiber membranes obtained from sample (a) pure PCL, sample (b) PCL/CNMIOPs prepared at pH 7.5 and washed with ethanol, and sample (c) PCL/CNMIOPs prepared at pH 7.5 and washed with water.

Sample	Diameter (nm)	*Ra* (nm)	*Rq* (nm)
a	256 ± 106	17.7 ± 10.4	20.8 ± 12.9
b	314 ± 118	56.6 ± 11.5	64.7 ± 12.4
c	352 ± 147	60.1 ± 22.0	71.2 ± 23.1

## Data Availability

All data generated or analyzed during this study are included in this published article.

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
