# Peer review of "Quantifying the Structure and Properties of Nanomagnetic Iron Oxide Particles for Enhanced Functionality through Chemical Synthesis"

_nanomaterials, 2023, doi:10.3390/nano13152242_

Round 1

Reviewer 1 Report

Dear Authors,

I have read your manuscript and my comments are in the attached file.

Best regards

Author Response

21th July 2023

Dear Editor,

We attach a revised version of the manuscript entitled “Quantifying the Structure and Properties of Nanomagnetic Iron Oxide Particles for Enhanced Functionality through Chemical Synthesis (Manuscript ID: nanomaterials-2524965).

All the comments were fair, constructive, and insightful. The comments raised by the reviewers have been taken into consideration. We appreciate this chance to revise our manuscript carefully and to carry out our changes accordingly. We also responded point-by-point to the reviewer’s comments as listed below.

I look forward to your answer.

Yours sincerely,

Johar Amin Ahmed Abdullah and Antonio Guerrero

Reviewers’ comments:

Reviewer: 1

Comments:

The paper evidences the impact of pH levels and washing conditions (different solvents) on the properties of nanomagnetic iron oxide particles. The iron oxide particles, prepared through a chemical method, were investigated by various characterization techniques (FTIR, SEM, TEM, XRD, ZSP, and VSM). Moreover, the application of the synthesized iron oxide particles in electrospun membranes was investigated.

This study contains a lot of information, the figures and tables are accompanied by detailed discussions, the references are appropriate to the subject addressed in the article. 42% of the total cited papers were published in the last 5 years. The paper can be published in Nanomaterials journal after some clarifications:

The authors gratefully acknowledge the reviewer’s comments and suggestions. All of them have been taken into consideration to improve the quality of the manuscript.

  1. line 231 – „...from 21.6 to 30.4 nm, showing a decreasing trend as the pH decreased” – As indicated Table 1, the average diameter decreases when pH value increases. Please, correct the sentence.

We greatly appreciate this observation. Accordingly, we have taken into account the suggestion to correct the statement by replacing "pH decreased" with "pH increased" (Lines 236).

  1. XRD - two phases are identified: alpha-Fe2O3 and magnetite (but not also maghemite gamma-Fe2O3) according to several JCPDS card numbers:

00-900-9782

00-210-1167

00-210-8027

00-153-2800

00-152-6955

00-900-5841

00-230-0616

00-230-0617

00-101-1240

00-722-8110

00-900-0139

Since it is difficult to identify each type of structure according to the JCPDS cards, I recommend to emphasize the structure for each individual file.

Why is not a single file identified for each type of structure? For example, four of them (00-900-9782, 00-210-1167, 00-101-1240, 00-900-0139) correspond to hematite R-3c.

We sincerely appreciate the valuable observations and comments provided by the reviewer. Based on our analysis using the identified JCPDS cards, we did not find any evidence of the maghemite phase. The JCPDS cards confirmed the presence of magnetite and hematite, as indicated in Table 3 and Figure 5. In response to the reviewer's recommendation, we have considered separating the JCPDS files for each phase to accurately identify the structures reported in Table 3 (Lines 268-281). Additionally, we have included the different space groups for each entry in Table 3 (Lines 265-266).

Furthermore, we acknowledge the error in reporting the crystal system of the sample prepared at pH 7.5 and washed with water. The correct percentages are 1.1% monoclinic and 16.3% trigonal-hexagonal axis, which have been updated in both the text and Figure 5.

In the IR spectra and magnetic measurements sections, the presence of maghemite is mentioned. How does it correlate with the XRD section?

Regarding the IR analysis, the peaks observed at 739-641 cm-1 were attributed to maghemite based on previous studies indicating similar bands in the IR spectra during the oxidation of magnetite during synthesis. However, the absence of maghemite peaks in the XRD analysis can be attributed to several factors, including their presence at very low concentrations and/or in an undetectable amorphous form. To address this, we have included a statement in the XRD section (Lines 301-305) justifying this limitation and noting that all samples exhibited crystallinity higher than 80%: “This limitation may suggest the absence of maghemite (γ–Fe2O3) peaks in XRD analysis. Furthermore, the absence of maghemite peaks in the XRD analysis could be attributed to either a lower concentration or the presence of an undetectable amorphous form. It is important to note that all samples exhibited crystallinity higher than 80%.”

We would like to correct the mention of the maghemite phase in the Magnetic properties section to reflect the presence of hematite instead (Line 343): “…exhibited mixed phases of magnetite and hematite, resulting in a Ms of 57.5 emu/g”.

According to reference [28], the monoclinic structure of magnetite is present at low temperatures (90 K). How would the presence of monoclinic phase be explained in the case of the current study?

We appreciate the query regarding the formation of the monoclinic phase at low temperatures. The presence of the monoclinic phase can be influenced by various factors, including the parameters evaluated in this study. However, it is worth noting that, in accordance with the reference [28], the percentage of monoclinic magnetite observed in our study was approximately 1% and 1.1%.

Thank you again to the reviewer for their valuable input, which has helped us improve the accuracy and clarity of our manuscript.

  1. Some typos:

- line 324 – „...as t pH and washing process ...” Please, correct the sentence.

- line 328 – „even with weak dipolar interactions he solvent”? Please, correct the sentence.

The authors apologize for any grammatical or other errors in the manuscript. Accordingly, thorough revisions and corrections have been made to the manuscript.

Reviewer 2 Report

This comprehensive study investigates the properties of nanomagnetic iron oxide particles (CNMIOPs) synthesized through a chemical method. The results show that pH and washing process significantly affect CNMIOPs' properties. Higher pH levels result in smaller particles with higher crystallinity and reduced crystalline anisotropy. SEM and TEM analysis confirm different morphologies, including cubic, spherical, and elongated shapes. Ethanol-washed CNMIOPs exhibit superior magnetic behavior, with the highest magnetization saturation at pH 12.5 (Ms = 58.3 emu/g). The stability of CNMIOPs ranges from -14.7 to -23.8 mV, and higher pH levels exhibit promising antioxidant activity.

Furthermore, the study explores the effects of pH and washing solvents on CNMIOPs-infused nanofiber membranes, with better dispersion observed with ethanol washing. Overall, this research provides valuable insights into the properties and behavior of CNMIOPs under varying pH and washing conditions.

Overall, the study appears to be well-structured and provides comprehensive information about the synthesis and properties of CNMIOPs. The use of various characterization techniques adds credibility to the findings. However, there are some areas where additional clarity or details could enhance the study:

  1. 1. Consider providing more information about the chemical synthesis method used to create the CNMIOPs. This would help readers better understand the processes involved.

  2. 2. Elaborate on the targeted application of PCL electrospun materials infused with CNMIOPs. What specific application is being considered, and how do the properties of CNMIOPs affect the performance in this application?

  3. 3. Provide more context or explanation regarding the significance of the stability and antioxidant activity of CNMIOPs under different pH levels. How do these findings contribute to the potential applications of these nanoparticles?

  4. 4. Include details about the potential limitations of the study and how they might impact the interpretation of the results.

  5. 5. If applicable, discuss any potential future directions or applications that could be explored based on these findings.

Author Response

21th July 2023

Dear Editor,

We attach a revised version of the manuscript entitled “Quantifying the Structure and Properties of Nanomagnetic Iron Oxide Particles for Enhanced Functionality through Chemical Synthesis (Manuscript ID: nanomaterials-2524965).

All the comments were fair, constructive, and insightful. The comments raised by the reviewers have been taken into consideration. We appreciate this chance to revise our manuscript carefully and to carry out our changes accordingly. We also responded point-by-point to the reviewer’s comments as listed below.

I look forward to your answer.

Yours sincerely,

Johar Amin Ahmed Abdullah and Antonio Guerrero

Reviewers’ comments:

Reviewer: 2

Comments:

This comprehensive study investigates the properties of nanomagnetic iron oxide particles (CNMIOPs) synthesized through a chemical method. The results show that pH and washing process significantly affect CNMIOPs' properties. Higher pH levels result in smaller particles with higher crystallinity and reduced crystalline anisotropy. SEM and TEM analysis confirm different morphologies, including cubic, spherical, and elongated shapes. Ethanol-washed CNMIOPs exhibit superior magnetic behavior, with the highest magnetization saturation at pH 12.5 (Ms = 58.3 emu/g). The stability of CNMIOPs ranges from -14.7 to -23.8 mV, and higher pH levels exhibit promising antioxidant activity.

Furthermore, the study explores the effects of pH and washing solvents on CNMIOPs-infused nanofiber membranes, with better dispersion observed with ethanol washing. Overall, this research provides valuable insights into the properties and behavior of CNMIOPs under varying pH and washing conditions.

Overall, the study appears to be well-structured and provides comprehensive information about the synthesis and properties of CNMIOPs. The use of various characterization techniques adds credibility to the findings. However, there are some areas where additional clarity or details could enhance the study:

The authors gratefully acknowledge the reviewer’s comments and suggestions. All of them have been taken into consideration to improve the quality of the manuscript.

  1. Consider providing more information about the chemical synthesis method used to create the CNMIOPs. This would help readers better understand the processes involved.

This comment from the reviewer is highly valuable, and we sincerely appreciate it. In response, we have included additional information about the synthesis process for creating CNMIOPs (Lines 105-112): ” This drying process facilitated the removal of excess moisture and ensured proper sample preparation. Subsequently, a final heat treatment was carried out at 500 °C for 5 hours. This high-temperature treatment was employed to eliminate any remaining excess matter and impurities, resulting in the isolation of the nanoparticles for subsequent characterization. It is worth noting that this two-step nanoparticle treatment involved intense heat transmission through a regulated drying oven and a muffle, ensuring efficient drying, sterilization, and removal of organic substances and impurities.”

  1. Elaborate on the targeted application of PCL electrospun materials infused with CNMIOPs. What specific application is being considered, and how do the properties of CNMIOPs affect the performance in this application?

We appreciate the reviewer's query. In response to this comment, we have taken into consideration the elaboration of potential applications for PCL electrospun materials infused with enhanced magnetic CNMIOPs in different parts of the manuscript. Firstly, in Lines 520-531 “The enhanced properties of CNMIOPs, including their magnetic properties, antioxidant activity, good dispersity, and ability to increase surface roughness, contribute to their potential applications in various fields.

One potential application of PCL electrospun materials infused with CNMIOPs is tissue engineering. The magnetic properties of CNMIOPs can enable remote manipulation and guidance of the scaffold within the body using external magnetic fields. This feature holds promise for applications such as targeted drug delivery and tissue regeneration. The incorporation of CNMIOPs into the electrospun PCL matrix can enhance the overall mechanical properties of the scaffold, improving its structural integrity and providing a suitable microenvironment for cell attachment, proliferation, and differentiation [13].” Furthermore, in Lines 550-558: “CNMIOPs prepared in this study displayed enhanced unique properties such as magnetic response, antioxidant activity, good dispersity, and surface roughness enhancement, making them highly versatile for various applications. In the context of tissue engineering, the incorporation of CNMIOPs into PCL electrospun materials offers promising potential. The magnetic properties enable remote manipulation and targeted drug delivery while improving the scaffold's mechanical integrity. These advancements create an optimal microenvironment for cell attachment, proliferation, and differentiation. Thus, the combination of CNMIOPs with PCL electrospun materials holds great promise for tissue engineering applications.”

  1. Provide more context or explanation regarding the significance of the stability and antioxidant activity of CNMIOPs under different pH levels. How do these findings contribute to the potential applications of these nanoparticles?

We appreciate the reviewer's query regarding the significance of the stability and antioxidant activity of CNMIOPs under different pH levels and their contribution to the potential applications of these nanoparticles. Accordingly, we have provided more context as below:

  • Stability (Lines 405-416):

“The stability of CNMIOPs prepared under different pH levels and washing processes is crucial for ensuring their robustness and suitability across various applications. The findings presented in this study demonstrate the ability of CNMIOPs to maintain their structural integrity and functionality in neutral environments. Nevertheless, further studies are needed to explore the stability of CNMIOPs across a broader range of pH levels, including acidic and alkaline conditions. This stability is particularly significant in applications where nanoparticles may encounter diverse pH environments, such as drug delivery systems or biomedical applications. The results suggest that CNMIOPs exhibit the capacity to maintain their stability and performance in water, ensuring their reliability and effectiveness in targeted applications. Furthermore, their performance can be enhanced through an extensive exploration of the parameters and conditions of the synthesis.”

  • Antioxidant activity (Lines 432-443):

“The antioxidant activity of CNMIOPs prepared under different pH levels and washing solvents is an essential attribute that enhances their potential applications in various fields, particularly in biomedical and therapeutic settings. Antioxidants play a vital role in reducing oxidative stress and neutralizing harmful free radicals, which are associated with various diseases and cellular damage. However, it should be noted that the ability of CNMIOPs to maintain their antioxidant activity across different conditions is limited. Nonetheless, their potential for combating oxidative stress in diverse physiological environments can be enhanced through various settings, including those explored in this study. This finding suggests that there is room for further improvement and utilization of CNMIOPs in formulations targeting oxidative stress-related diseases or in systems where antioxidant properties are desired, such as tissue engineering scaffolds.”

  1. Include details about the potential limitations of the study and how they might impact the interpretation of the results.

We appreciate the reviewer's suggestion to address the limitations of the study for a comprehensive understanding of the findings. Accordingly, we have included a discussion of these limitations in the manuscript, specifically in Lines 559-570: “However, it is important to acknowledge potential limitations in the study, including the specific conditions and characterization techniques used, which may limit the generalizability of the findings. Additionally, further exploration of factors influencing the properties of the nanoparticles is needed. To enhance our understanding, future research should focus on examining the biocompatibility, toxicity, and long-term storage effects of CNMIOPs, particularly for biomedical applications. Investigating the influence of factors such as temperature, reaction time, and reactant concentration on the properties of CNMIOPs is crucial for a comprehensive understanding. Moreover, it would be highly beneficial to explore green methods or combine different approaches to enhance the diverse, unique properties of CNMIOPs and scale up their synthesis for industrial applications. By considering these limitations and focusing on these research areas, we can advance our understanding of CNMIOPs and unlock their full potential for various applications.”

  1. If applicable, discuss any potential future directions or applications that could be explored based on these findings.

We appreciate the reviewer's suggestion to discuss potential future directions or applications based on our findings. As mentioned in the previous comment, we have addressed this suggestion and provided details on potential future directions.

Round 2

Reviewer 2 Report

The author has revised it according to my suggestion. I recommend that the manuscript can be accepted and published.